# Assessment of potential risk factors for COVID-19 among health care workers in a health care setting in Delhi, India -a cohort study

Mridu Dudeja[1☯], Aqsa Shaikh[2☯], Farzana Islam[2☯], Yasir Alvi[2☯]*, Mohammad Ahmad[3], Varun Kashyap[2‡], Vishal Singh[4‡], Anisur Rahman[3], Meely Panda[5], Neetu Shree[1], Shyamasree Nandy[2], Vineet Jain[6]

1 Department of Microbiology, Hamdard Institute of Medical Sciences and Research, New Delhi, India, 2 Department of Community Medicine, Hamdard Institute of Medical Sciences and Research, New Delhi, India, 3 World Health Organization, Country Office, New Delhi, India, 4 Zonal AEFI Coordinator, Ministry of Health and Family Welfare, Government of India, New Delhi, India, 5 Department of Community and Family Medicine, All India Institute of Medical Science, Telangana, India, 6 Department of Medicine, Hamdard Institute of Medical Sciences and Research, New Delhi, India

☯ These authors contributed equally to this work.
‡ VK and VS also contributed equally to this work.
* yasiralvi13@gmail.com

**Data Availability Statement:** The research data are available at public repository Zenodo. https://doi.org/10.5281/zenodo.5703338 https://zenodo.org/record/5703338#.YhqZRi8RpQI.

## Abstract

### Introduction

Healthcare workers (HCW) are most vulnerable to contracting COVID-19 infection. Understanding the extent of human-to-human transmission of the COVID-19 infection among HCWs is critical in managing this infection and for policy making. We did this study to estimate new infection by seroconversion among HCWs in recent contact with COVID-19 and predict the risk factors for infection.

### Methods

A cohort study was conducted at a tertiary care COVID-19 hospital in New Delhi during the first and second waves of the COVID-19 pandemic. All HCWs working in the hospital during the study period who came in recent contact with the patients were our study population. The data was collected by a detailed face-to-face interview, serological assessment for anti-COVID-19 antibodies at baseline and end line, and daily symptoms. Potential risk factors for seroprevalence and seroconversion were analyzed by logistic regression keeping the significance at p<0.05.

### Results

A total of 192 HCWs were recruited in this study, out of which 119 (62.0%) were seropositive. Almost all were wearing Personal protective equipment (PPE) and following Infection prevention and control (IPC) measures during their recent contact with a COVID-19 patient. Seroconversion was observed among 36.7% of HCWs, while 64.0% had a serial rise in the titer of antibodies during the follow-up period. Seropositivity was negatively associated with

**Funding:** This study was funded by World Health Organization, under the UNITY Studies. The study protocol was based on pre-design WHO UNITY protocol, adapted for local settings. The sponsor did not play any role in data collection and analysis, decision to publish, or preparation of the manuscript. https://www.who.int/emergencies/diseases/novel-coronavirus-2019/technical-guidance/early-investigations.

**Competing interests:** NO authors have competing interests.

being a doctor (odds ratio [OR] 0.35, 95% Confidence Interval [CI] 0.18–0.71), having COVID-19 symptoms (OR 0.21, 95% CI 0.05–0.82), having comorbidities (OR 0.14, 95% CI 0.03–0.67), and received IPC training (OR 0.25, 95% CI 0.07–0.86), while positively associated with partial (OR 3.30, 95% CI 1.26–8.69), as well as complete vaccination for COVID-19 (OR 2.43, 95% CI 1.12–5.27). Seroconversion was positively associated with doctor as a profession (OR 13.04, 95% CI 3.39–50.25) and with partially (OR 4.35, 95% CI 1.07–17.65), as well as fully vaccinated for COVID-19 (OR 6.08, 95% CI 1.73–21.4). No significant association was observed between adherence to any IPC measures and PPE adopted by the HCW during the recent contact with COVID-19 patients and seroconversion.

## Conclusion

Almost all the HCW practiced IPC measures in these settings. High seropositivity and seroconversion are most likely due to concurrent vaccination against COVID-19 rather than recent exposure to COVID-19 patients. Further studies using anti-N antibodies serology may help us find the reason for the seropositivity and seroconversion among HCWs.

## Introduction

The SARS-CoV-2 virus, a member of *the Coronaviridae* virus family, is the agent of the COVID-19 infection. This virus has constantly been changing and becoming diverse, and this property has helped this virus to spread among vulnerable populations and jeopardize any healthcare system quickly. The SARS-CoV-2 virus transmits from one host to another via respiratory droplets, aerosol, contact with bodily fluids, and contaminated surfaces [1]. Asymptomatic people may be able to transmit infection, while individuals who have not reported proximity to any known case have also been infected [2, 3]. India has the largest number of confirmed cases of COVID -19 in Asia and the second-highest number of confirmed cases in the world [4]. The national capital, New Delhi, has been one of the earliest and most affected cities in the Indian epidemic. The test and treat policy was the backbone of Delhi's fight against COVID-19. Contrary to the first wave, which observed a low number of COVID-19 cases plateauing over time, the second wave saw an increase in cases and deaths, waning and waxing off after a sudden short and high peak [5].

Healthcare workers (HCWs) were at the center of the COVID -19 crisis. From line listing, diagnosis, treatment, rehabilitation, and home visits to prevention services like vaccination or quarantine, all required the health workforce to act as front liners. Adding to these challenges were financial insecurities, violence against healthcare workers, the wrath of families affected, and governmental policy backlash. HCWs providing COVID-19 care were at increased risk of acquiring infection because of their situation. With the scarcity of the workforce during the pandemic on top of an existing shortage of healthcare workers, the vulnerability of HCWs to contracting COVID-19 makes them needier for research to understand the transmission dynamics. Furthermore, they can also be a potential source of hospital-acquired infection. COVID-19 infection was common among nursing staff, while death was seen more among doctors, with the highest case fatality rate seen in the age group over 70 years [6]. They also have a role in implementing adequate infection prevention and control (IPC) measures and the use of Personal protective equipment (PPE) in healthcare facilities. Several advisories and directives have been issued by the Government of India's Ministry of Health and Family, India for managing this crisis among Health care workers. They stressed activating hospital infection

control committees, identifying nodal officers to respond to Health Associated Infections, and following updated guidelines.

There are a handful of research in other countries on the serology of COVID-19 among HCWs. The seropositivity among HCW varies from 3.3% in China, 9.3% in Spain, 10.1% in the United Kingdom (UK), and 6.35–33% in the United States (US), while it was found to be 8.7% in a metanalysis [7–12]. Some studies have found a seroconversion from 0.77% in the UK to 77.8% in Germany over a different period [7, 13]. However, these serological data among Indian HCWs is limited. A countrywide serosurvey after the first wave observed a seroprevalence among HCWs at 25.6%, while it was 12.1% among HCWs from a teaching hospital in Delhi [14, 15]. Investigating the serological response and assessing the potential risk factors among health workers may help characterize virus transmission patterns, prevent future infections of health workers, and prevent the healthcare-associated spread of COVID-19. Thus, this study aimed to understand the extent of human-to-human transmission, determine the risk factors for COVID-19 among Indian HCWs and evaluate the effectiveness of IPC measures among them.

## Material and methods

### Study design and population

This prospective cohort study was carried out among HCWs between December 2020 to June 2021, during which India faced its first and second wave of the COVID-19 pandemic. The first wave hit India in March 2020, with a peak in mid-September and extended till December 2020, while the second deadlier wave was observed between March to July 2021 [5, 16]. The study was conducted at Hamdard Institute of Medical Sciences and Research (HIMSR) and Hakeem Abdul Hameed Centenary Hospital, a dedicated COVID-19 Hospital of 200 beds in South East Delhi, India. The study population included all the health personnel working in this hospital who had come in contact with or had been exposed recently (within 72 hours) to a COVID-19 patient receiving care. This hospital had 1050 healthcare workers involved in COVID-19 care, and details of study participants and their recruitment can be seen in Fig 1.

**Inclusion criteria.** All HCWs exposed to COVID-19 patients receiving care in this healthcare facility within 72 hours of confirmation of the diagnosis with

- Close contact (within 1 meter) with a laboratory-confirmed case, or

- Exposure to patient's blood or body fluids, or

- Exposure to patient's used materials, devices, or equipment, or

- Exposure to environmental surfaces around the confirmed case, including their bed, table, wheelchair, ward corridor, etc.

**Exclusion criteria.** • HCWs who also worked in another healthcare facility

- HCWs who had already contracted or were positive or had a confirmed COVID-19 case among their household/close contacts.

- HCWs who were clinically serious and could not participate in the study.

### Sample size

Based on an extensive review of the literature [8, 17], in Indian settings, we estimated the proportion of seroconversion in the general population at 25% and the proportion of seroconversion in healthcare workers at 36%. Taking significance at 5% and power at 80%, we calculated a sample size of 130, which was adjusted considering the attrition rate of 25% to 170 (S1 File).

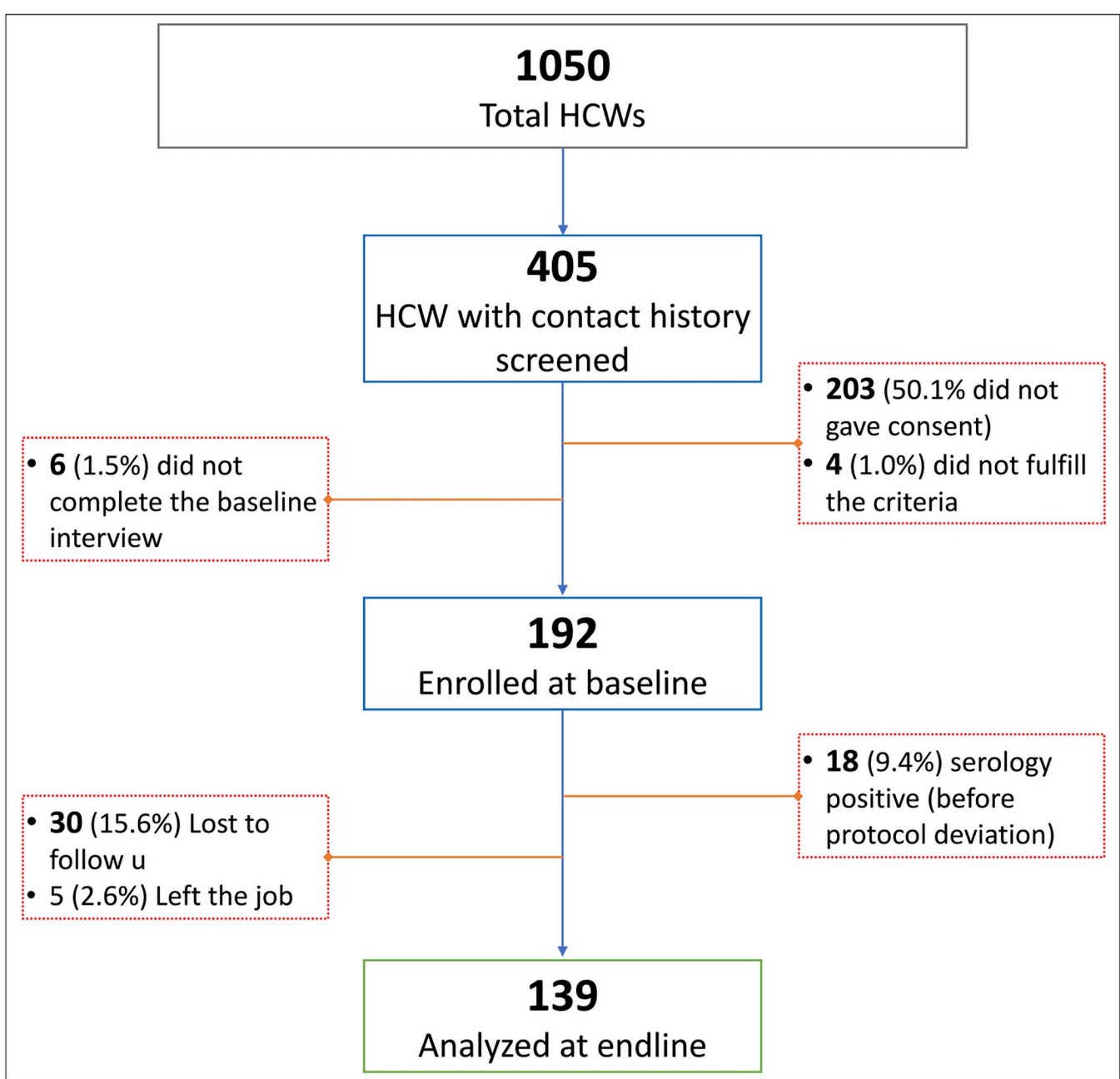

**Fig 1. Description of enrolment of study participants.**

## Operational definitions of study outcomes

**Seropositivity.** Study participants who were positive for SARS-CoV-2 antibodies detected with WANTAI SARS-CoV-2 Ab (Enzyme-linked immunosorbent assay) ELISA at baseline, due to prior COVID- 19 infection, or vaccination [absorbance (A)/ Cut-off value (C.O) $\geq 1$].

**Seroconversion.** Study participants who were seronegative at baseline but became positive at the endline.

**Seroconversion rate and secondary infection rate.** Seroconversion rate is a ratio of the HCWs who were enrolled and confirmed for new infections of COVID-19 assessed through

serological assays on paired samples, divided by susceptible contacts (total enrolled HCWs). Due to the limitation of our study, we consider this a proxy for the Secondary infection rate.

**The serial rise in titer.** Those HCWs who were positive in the endline serum sample and had either increased titer from the baseline or had the maximum possible value.

**Incubation period.** The duration between the current exposure to COVID-19 patient and the beginning of the symptoms of the disease.

## Participant recruitment

When a confirmed case of COVID-19 was admitted or detected, all its suspected contacts were traced by the Covid Surveillance Unit. Potential participants with recent contact were screened, and those eligible were invited to participate in the study and a Patient Information Sheet explaining the study was provided. Self-reporting of contact and referrals were also utilized for enrolment. Those who gave written informed consent were included and interviewed with a baseline questionnaire and first blood collection (Fig 1). They were monitored daily by a self-administered symptoms diary for their symptoms along with regular telephonic contact during the 21 day follow-up period. During 22–28 days from the first visit, the study participants were requested to visit the study site for endline data collection and a second serum sample along with submission of the symptoms diary, details of which are given in the S1 File. Before entering data, all the participants were given an anonymized unique ID so that none apart from the investigators could know the participant's details. The hospital offered the RTPCT test to all the participants who needed it, and those with poor IPC practices were referred for refresher IPC training.

## Serological assessment

During baseline and endline visits, two milliliters of blood were collected, coded with the participant's unique ID, and anonymized. The first sample collected after enrolment was considered the baseline blood sample. All the subjects were recalled after 21 days (from the date of baseline sample collection) for the collection of endline blood samples. (*Protocol deviation 1*: In the initial proposal, we had planned to call only those seronegative at baseline; later, after protocol deviation, all HCWs were called for endline assessment). Anti-SARS-CoV-2-total antibody detection was done by the Wantai SARS-CoV-2-Ab ELISA kit, which detects total antibodies against the SARS-CoV-2 virus [18]. The cut-off value (C.O.) was calculated based on /the absorbance value (AV) for three negative calibrators. Those samples with AV less than the C.O. were reported as negative results (AV / C.O. < 1). Specimens with AV equal to or greater than the C.O. were considered positive. After decoding, the study participants were informed about their baseline and endline antibody results, and counseling was done accordingly (S1 File).

## Statistical analyses

The data were collected, entered, and managed in Microsoft excel with appropriate coding. Participant information was anonymized using a unique code accessible only to investigators. The data entry operator and investigators regularly checked for the accuracy and consistency of the questionnaires and did double data entry and appropriate cleaning to prevent any possible errors. For statistical analysis, the data was exported, and IBM SPSS version 26 (SPSS Inc., Armonk, NY) was used. The categorical variables were presented as percentages (%), while for continuous normally distributed variables, we calculated the mean with standard deviation and median with interquartile range for non-normally distributed variables. Association between various factors and outcome of interest was done by Pearson's chi-square and binary

logistic regression, using IBM SPSS. All tests are performed at a 5% significance level; thus, the p-value < 0.05 was taken as a significant association.

### Ethical considerations

Ethical considerations for the study were given the utmost care, and all norms of confidentiality, autonomy, beneficence, and consent were followed. The study started after approval by the Research Project Advisory Committee and Institutional Ethics committee of Hamdard Institute of Medical Science and Research, New Delhi (IEC-2020/14). Additionally, the required approvals from the district administration, hospital administration, and medical college were received. Written informed consent in English / Hindi from each participant was obtained. All the project staff was trained in Good Clinical Practice. Guidelines of International Conference on harmonization–Good clinical practices were followed as appropriate. Appropriate referral for clinical symptoms, RTPCR testing, and refresher IPC training was done as required.

### Results

A total of 405 HCWs who were potential participants due to recent exposure were screened, and 192 were enrolled (participation rate 48.6%). They were interviewed, and the baseline blood sample was collected. Of them, 139 (72.4%) were also interviewed at end-line with a second serology assessment (Fig 1).

The sociodemographic profile, working conditions, comorbidities, and other characteristics of the study participants are highlighted in Table 1. Most participants were of the young age group, with a mean age of 31.7 ±9.3 years and an almost equal distribution across gender categories. More than half of the participants were paramedics, and a quarter were doctors and nurses. The lowest percentage of unvaccinated individuals was among doctors (N = 24, 47.1%), while paramedics had the lowest rate (N = 10, 9.8%) of fully vaccinated individuals (p<0.001) (Fig 2). All participants had taken Covishield (AstraZeneca/ChAdOx1nCoV- 19) vaccine. We found an incubation period of 7.9 ± 6.8 (median = 7.5) days in our study population, as shown in Table 1.

About two-thirds (63.5%) of the HCW had close contact exposure with COVID-19 patients; among them, 27 (14.1%) had a prolonged face-to-face exposure, and 10 (5.2%) had exposure during the aerosol-generating procedure, while exposure with body fluid was observed in 13 (6.8%). Exposure to patient material was seen in 58 (30.2%), while exposure to the surface around the patient was noted in 164 (85.4%). Table 2 also shows various exposure to COVID-19 patients across their professions. Most nurses and doctors had direct contact exposure.

The usage of various PPE during recent contact with COVID- 19 patients is shown in S2 Fig. We observed a trend in adherence to PPE, with almost all participants following the PPE protocol during the high-risk procedure and fewer participants during direct face-to-face contact. In addition, nearly three-fourths of the HCWs wore face masks, while face shields and protective eyewear were used by only one-fourth during prolonged face-to-face exposure. We also observed that most participants wore all the PPE required during the Aerosol generating Procedure and body fluids exposure. Adherence to PPE statistically differs among health workers, as shown in Table 3. The nurses used PPE the most during the recent close contact with the COVID-19 case, while paramedics were the least adherent (p<0.01). Hand hygiene practice is also shown in S1 Table, and nurses were reported to have the best rates.

Our study did not observe any association between exposure type, adherence to PPE and IPC practices with seroconversion, and serial rise in titre of antibodies against COVID-19 between 3–4 weeks of contact (Tables 4 and 5).

**Table 1. Healthcare workers characteristics at baseline and endline.**

| Variables | | Frequency (Percentage) | |
|---|---|---|---|
| | | Baseline (N = 192) | End line (N = 139) |
| **Gender** | Female | 90 (46.9%) | 64 (46.0%) |
| | Male | 102 (53.1%) | 75 (54.0%) |
| **Age (years)** | 18 to 30 | 116 (60.4%) | 80 (57.6%) |
| | 31 to 60 | 75 (39.1%) | 58 (41.7%) |
| | more than 61 | 1 (0.5%) | 1 (0.7%) |
| | Mean ± SD | 31.71±9.27 | 31.95±9.58 |
| **Category** | Doctor | 51 (26.6%) | 35 (25.2%) |
| | Nurse | 39 (20.3%) | 26 (18.7%) |
| | Paramedic | 102 (53.1%) | 78 (56.1%) |
| **Area of work** | High risk | 86 (44.8%) | 58 (41.7%) |
| | Low risk | 106 (55.2%) | 81 (58.3%) |
| **Smoking (current)** | Yes | 24 (12.5%) | 19 (13.7%) |
| | No | 168 (87.5%) | 120 (86.3%) |
| **Comorbidities** | Obesity | 4 (2.1%) | 3 (2.2%) |
| | Diabetes | 1 (0.5%) | 1 (0.7%) |
| | Chronic Lung Disease | 1 (0.5%) | 1 (0.7%) |
| | Other Comorbidities | 6 (3.1%) | 4 (2.9%) |
| **Vaccination against Covid-19** | Not Vaccinated | 121 (63.0%) | 77 (55.4%) |
| | Partially Vaccinated | 29 (15.1%) | 26 (18.7%) |
| | Vaccinated | 42 (21.9%) | 36 (25.9%) |
| **Recent IPC training** | Not received | 26 (13.5%) | 21 (15.1%) |
| | Less than 2 hr | 144 (75.0%) | 106 (76.3%) |
| | More than 2 hr | 22 (11.5%) | 12 (8.6%) |
| **Incubation period** | Median (IQR) | | 7.5 (1.0–15.0) |

*Percentages may not total 100 because of rounding. SD denotes Standard deviation, IQR: Interquartile range, hr: hours, IPC: infection prevention control

†Gender was reported by the participants

Table 6 shows the prevalence of seropositivity, seroconversion, and a serial rise in antibody titer between the baseline and endline of the study population along with their 95% confidence intervals.

We observed a seropositivity of 62% (95% CI 54.9–68.6) among our participants. We also observed high seropositivity among participants with various characteristics, including higher age group (73.3%), Paramedic staff (71.6%), working in the low-risk area (67.9%), vaccinated with Covid-19 (73.8%) and not received Infection prevention control training (76.9%). We observed that more than one-third (36.7%) of the HCW became seroconverted. In terms of a rise in titer, it was observed in 64.0% of HCW. The seroconversion rate was 63.2% among doctors, 42.9% among nurses, and 13.0% among paramedical staff. Doctors' antibody titer was observed to increase the most (71.4%), while among nurses, it was seen at 53.8%.

In our study, only 5.7% had symptoms after recent contact with the COVID-19 patients admitted to the health care settings, while 7.2% had during their follow-up period. Most symptomatic HCWs did not develop antibodies at the endline. The most common symptoms were headache and fatigue (S2 Table).

We also assessed the risk of seropositivity, seroconversion, and rise in titer as highlighted in Table 7. The seropositivity was significantly and negatively associated with a doctor as profession (OR 0.35, 95% CI 0.18–0.71), COVID-19 symptoms (OR 0.21, 95% CI 0.05–0.82),

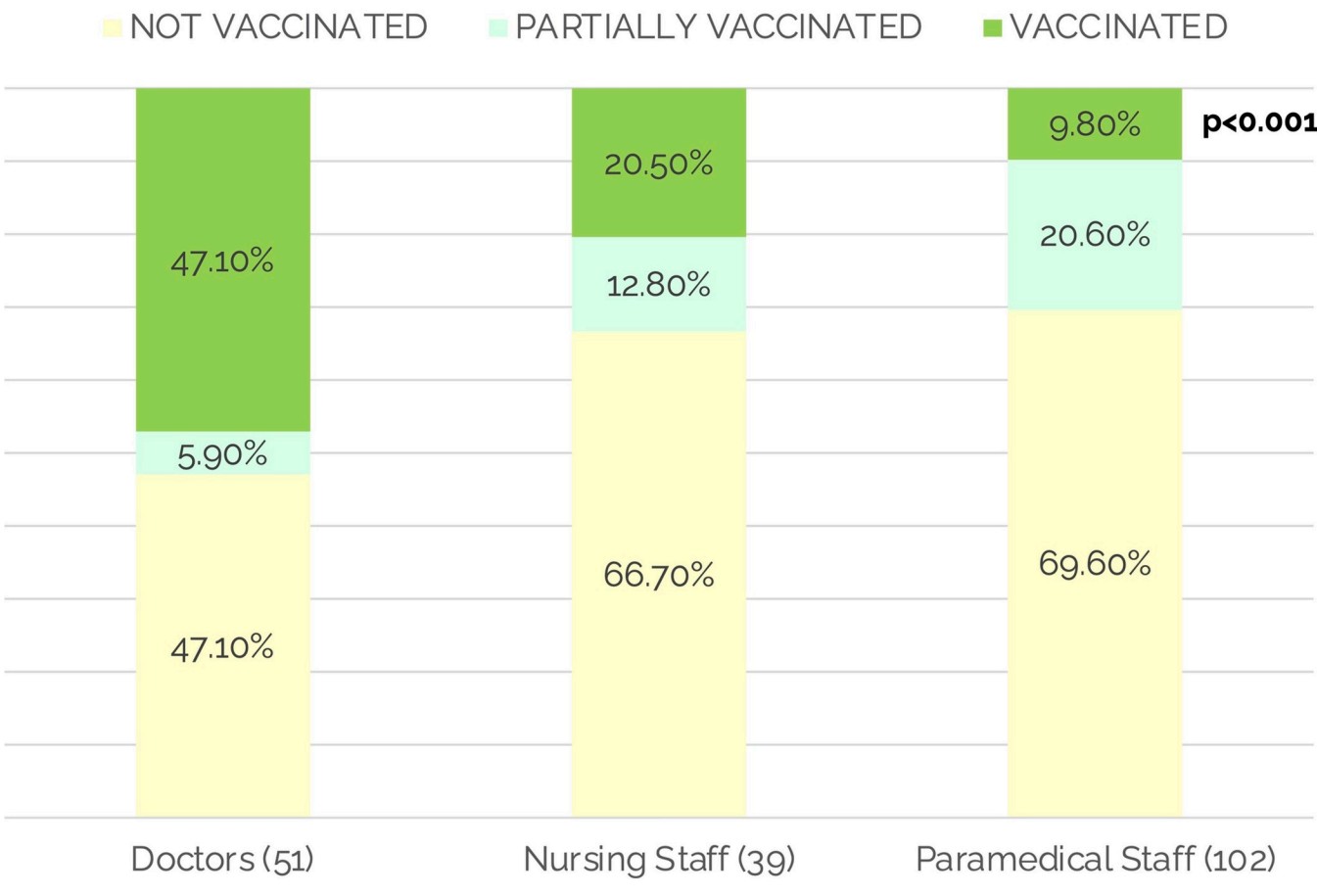

**Fig 2. Vaccination status among HCW at baseline.**

comorbidities (OR 0.14, 95% CI 0.03–0.67), and IPC training (OR 0.25, 95% CI 0.07–0.86), while positively associated with middle age (OR 2.23, 95% CI 1.19–4.19), partially (OR 3.30, 95% CI 1.26–8.69), as well as fully vaccinated for COVID-19 (OR 2.43, 95% CI 1.12–5.27). The seroconversion was observed to be significantly and positively associated with a doctor as a

**Table 2. Type of exposure among the study participants (N = 192).**

| Type of Exposure | Frequency (% among all HCW) | Doctors (n = 51) | Nursing Staff (n = 39) | Paramedical staff (n = 102) | P value |
|---|---|---|---|---|---|
| **Close contact exposure** | 122 (63.5%) | 35 (68.6%) | 31 (79.5%) | 56 (54.9%) | 0.017 |
| Prolonged face-to-face exposure | 27 (14.1%) | 8 (15.7%) | 7 (17.9%) | 12 (11.8%) | 0.975 |
| Exposure during aerosolizing procedures | 10 (5.2%) | 4 (7.8%) | 4 (10.3%) | 2 (2.0%) | 0.198 |
| Direct exposure with body fluid | 13(6.8%) | 5 (9.8%) | 5 (12.8%) | 3 (2.9%) | 0.171^ |
| **Patient's materials exposure** | 58 (30.2%) | 15 (29.4%) | 12 (30.8%) | 31 (30.4%) | 0.989 |
| Patient's body fluid via materials exposure | 4 (2.1%) | 3 (5.6%) | 1 (2.6%) | 0 (0%) | 0.016^ |
| **Surface exposure** | 164 (85.4%) | 44 (86.3%) | 34 (87.2%) | 86 (84.3%) | 0.691 |
| Patient's body fluid via surface around patient | 5 (2.6%) | 3 (5.9%) | 1 (2.6%) | 1 (1%) | 0.167^ |

*Percentages may not total 100 because of rounding. HCW denotes healthcare worker

^ Significance was assessed using Fisher exact test in these variables while in rest Chi-square test was used

**Table 3. Adherence to various PPE among doctors, nurses, and paramedics during recent contact with COVID-19 patient.**

| PPE | Any contact | Doctor | Nurse | Paramedics | p value^ |
|-----|-------------|--------|-------|------------|----------|
|     | N = 187* | N = 50 | N = 38 | N = 99 |          |
| **Medical/Surgical Mask** | 135 (72.2%) | 44 (88.0%) | 27 (71.1%) | 64 (64.6%) | p = 0.011 |
| **Respirator** | 152 (81.3%) | 39 (78.0%) | 37 (97.4%) | 76 (76.8%) | p = 0.017 |
| **Face Shield** | 49 (26.2%) | 14 (28.0%) | 18 (47.4%) | 17 (17.2%) | p = 0.001 |
| **Gloves** | 112 (59.9%) | 33 (66.0%) | 34 (89.5%) | 45 (45.5%) | p < 0.001 |
| **Goggles/Glasses** | 59 (31.6%) | 21 (42.0) | 19 (50.0%) | 19 (19.2%) | p < 0.001 |
| **Gown** | 98 (52.4%) | 27 (54.0%) | 36 (94.7%) | 35 (35.4%) | p < 0.001 |
| **Coverall** | 44 (23.5%) | 19 (38.0%) | 16 (42.1%) | 9 (9.1%) | p < 0.001 |
| **Head Cover** | 104 (55.6%) | 27 (54.0%) | 34 (89.5%) | 43 (43.4%) | p < 0.001 |
| **Shoe Cover** | 98 (52.4%) | 26 (52.0%) | 32 (84.2%) | 40 (40.4%) | p < 0.001 |

*5 HCW did not respond to this question

+ Percentages may not total 100 because of rounding off.

^ Significance was assessed using the Chi-square test

profession (OR 13.04, 95% CI 3.39–50.25) and with partially (OR 4.35, 95% CI 1.07–17.65), as well as fully vaccinated for COVID-19 (OR 6.08, 95% CI 1.73–21.4). The serial rise in titer was significantly and negatively associated with the serial rise in titer of antibodies with a history of symptoms (OR 0.19; 95% CI 0.06–0.64), smokers(OR 0.35, 95% CI 0.13–0.94), HCW with comorbidities (OR 0.08, 95% CI 0.01–0.71), recent full IPC Training (OR 0.07, 95% CI 0.01–

**Table 4. Exposure type along with IPC measures among the study participants and seroconversion.**

| Type of Exposure | | Seroconversion | | | Serial rise of titer | | |
|------------------|------|------|------|---------|------|------|---------|
|                  |      | No | Yes | p value | No | Yes | p value |
| **Close contact** | Yes | 25 (67.6%) | 12 (32.4%) | 0.32 | 38(44.2%) | 48(55.8%) | 0.010 |
|                   | No | 6 (50%) | 6 (50.0%) |      | 12(22.6%) | 41(77.4%) |      |
| **Prolonged face to face** | Yes | 8 (88.9%) | 1 (11.1%) | 0.22 | 11(57.9%) | 8 (42.1%) | 0.158 |
|                            | No | 17 (60.7%) | 11 (39.3%) |     | 27(39.7%) | 41(60.3%) |      |
| **Aerosol** | Yes | 5 (100.0%) | 0 (0.0%) | 0.152 | 5(100.0%) | 0 (0.0%) | 0.014 |
|             | No | 20 (62.5%) | 12 (37.5%) |      | 33(40.7%) | 48(59.3%) |      |
| **Patient fluid—direct** | Yes | 4 (100.0%) | 0 (0.0%) | 0.282 | 5 (62.5%) | 3 (37.5%) | 0.457 |
|                          | No | 21 (63.6%) | 12 (36.4%) |     | 33(42.3%) | 45(57.7%) |      |
| **Patient material** | Yes | 15 (88.2%) | 2 (11.8%) | 0.008 | 19(48.7%) | 20(51.3%) | 0.051 |
|                      | No | 16 (50.0%) | 16 (50.0%) |      | 31(31.0%) | 69(69.0%) |      |
| **Patient fluid–indirect@** | Yes | 1 (100.0%) | 0 (0.0%) | 1.000 | 1(100.0%) | 0 (0.0%) | 0.487 |
|                             | No | 14 (87.5%) | 2 (12.5%) |      | 18(47.4%) | 20(52.6%) |      |
| **Surface exposure** | Yes | 23 (60.5%) | 15 (39.5%) | 0.638 | 42(34.1%) | 81(65.9%) | 0.259 |
|                      | No | 7 (70.0%) | 3 (30.0%) |      | 7 (46.7%) | 8 (53.3%) |      |
|                      | Unknown | 1 (100.0%) | 0 (0.0%) |   | 1(100.0%) | 0 (0.0%) |      |
| **Body fluid via Surface** | Yes | 1 (100.0%) | 0 (0.0%) | 1.000 | 1(100.0%) | 0 (0.0%) | 0.341 |
|                            | No | 22 (59.5%) | 15 (40.5%) |     | 41(33.6%) | 81(66.4%) |      |

*Percentages may not total 100 because of rounding off. IPC denotes Infection prevention and control

# Prolonged face-to-face contact was considered when it was >15 minutes

^ Significance was assessed using Fisher's exact test in these variables while in the rest of the cases, the Chi-square test was used

@ exposure to patient's body fluid via socked patient's material

**Table 5. Adherence to PPE and seroconversion and serial rise of titre.**

| PPE used | | Seroconversion | | | Serial rise of titre | | |
|---|---|---|---|---|---|---|---|
| | | No | Yes | p value | No | Yes | p value |
| **Surgical Mask** | Yes | 24 (64.9%) | 13 (35.1%) | 0.738 | 35(36.5%) | 61(63.5%) | 0.989 |
| | No | 7 (58.3%) | 5 (41.7%) | | 15(36.6%) | 26(63.4%) | |
| **Respirator** | Yes | 25 (62.5%) | 15 (37.5%) | 1.000 | 40(35.7%) | 72(64.3%) | 0.819 |
| | No | 6 (66.7%) | 3 (33.3%) | | 10(40.0%) | 15(60.0%) | |
| **Face Shield** | Yes | 7 (87.5%) | 1 (12.5%) | 0.229 | 12(33.3%) | 24(66.7%) | 0.691 |
| | No | 24 (58.5%) | 17 (41.5%) | | 38(37.6%) | 63(62.4%) | |
| **Gloves** | Yes | 18 (64.3%) | 10 (35.7%) | 0.864 | 29(35.8%) | 52(64.2%) | 0.858 |
| | No | 13 (61.9%) | 8 (38.1%) | | 21(37.5%) | 35(62.5%) | |
| **Goggles/ Glasses** | Yes | 12 (66.7%) | 6 (33.3%) | 0.767 | 16(38.1%) | 26(61.9%) | 0.848 |
| | No | 19 (61.3%) | 12 (38.7%) | | 34(35.8%) | 61(64.2%) | |
| **Gown** | Yes | 15 (65.2%) | 8 (34.8%) | 0.790 | 26(37.1%) | 44(62.9%) | 0.872 |
| | No | 16 (61.5%) | 10 (38.5%) | | 24(35.8%) | 43(64.2%) | |
| **Coverall** | Yes | 5 (62.5%) | 3 (37.5%) | 0.961 | 10(34.5%) | 19(65.5%) | 0.800 |
| | No | 26 (63.4%) | 15 (36.6%) | | 40(37.0%) | 68(63.0%) | |
| **Head Cover** | Yes | 17 (73.9%) | 6 (26.1%) | 0.146 | 27(35.5%) | 49(64.5%) | 0.792 |
| | No | 14 (53.8%) | 12 (46.2%) | | 23(37.7%) | 38(62.3%) | |
| **Shoe Cover** | Yes | 17 (73.9%) | 6 (26.1%) | 0.146 | 25(35.2%) | 46(64.8%) | 0.746 |
| | No | 14 (53.8%) | 12 (46.2%) | | 25(37.9%) | 41(62.1%) | |
| **Remove gloves after contact** | yes | 18 (62.1%) | 11 (37.9%) | 0.834 | 33(35.1%) | 61(64.9%) | 0.759 |
| | no | 13 (65.0%) | 7 (35.0%) | | 17(37.8%) | 28(62.2%) | |
| **Hand hygiene performed** | adequate | 28 (63.6%) | 16 (36.4%) | 1.000 | 46(34.6%) | 87(65.4%) | 0.188 |
| | Not adequate | 3 (60.0%) | 2 (40.0%) | | 4 (66.7%) | 2 (33.3%) | |

*Percentages may not total 100 because of rounding off. PPE denotes personal protective equipment

^ Significance was assessed using Fisher's exact test in these variables, while in the rest of the cases Chi-square test was used

0.63), while positively associated with partially (OR 7.87, 95% CI 2.18–28.40), as well as fully vaccinated for COVID-19 (OR 3.59, 95% CI 1.46–8.87).

## Discussion

In this cohort study done at a tertiary care hospital in New Delhi, we found that seropositivity against COVID-19 was 62.0%, which was significantly and negatively associated with being a doctor, having symptoms, comorbidities, and recent IPC training while positively associated with being partially and fully vaccinated for COVID-19. This research supports the previous studies of higher seroprevalence of antibodies against COVID-19 among healthcare workers. These may be due to a higher transmission risk or the ongoing immunization program against COVID-19. We also observed seroconversion among 36.7%, while 64.0% had a serial rise in the titre of antibodies during our follow-up period. The seroconversion was higher in doctors and nurses (63.2% and 42.9%, respectively) compared to paramedics staff (13.0%). Seroconversion was positively associated with being a doctor and partially and fully vaccinated for COVID-19. We observed a negative and significant relationship between a serial rise in the titre of antibodies with recent symptoms suggestive of COVID-19, smoking, having comorbidities, and the recent IPC training, while positively associated with being partially and fully vaccinated for COVID-19. Adherence to the infection prevention measure adopted by the HCW during the recent contact with COVID-19 patients was not significantly associated with seroconversion or serial rise in titer.

**Table 6. Distribution of seroprevalence, seroconversion, and increase in antibody titer of HCW from baseline to endline among healthcare workers.**

| Variable | | Seroprevalence n = 192 | | Seroconversion n = 49 | | Serial rise in titre n = 139 | |
|---|---|---|---|---|---|---|---|
| | | N (%) | 95% CI | N (%) | 95% CI | N (%) | 95% CI |
| **Overall** | | 119 (62.0%) | 54.9–68.6 | 18 (36.7%) | 24.7–50.7 | 89 (64.0%) | 55.8–71.5 |
| **Profession** | Doctor | 24 (47.1%) | 34.1–60.5 | 12 (63.2%) | 41.0–80.9 | 25 (71.4%) | 54.9–83.7 |
| | Nurse | 22 (56.4%) | 41.0–70.7 | 3 (42.9%) | 15.8–75.0 | 14 (53.8%) | 35.5–71.2 |
| | Paramedic | 73 (71.6%) | 62.2–79.4 | 3 (13.0%) | 4.5–32.1 | 50 (64.1%) | 53.0–73.9 |
| **Gender** | Female | 52 (57.8%) | 47.5–67.5 | 10 (38.5%) | 22.4–57.5 | 38 (59.4%) | 47.1–70.5 |
| | Male | 67 (65.7%) | 56.1–74.2 | 8 (34.8%) | 18.8–55.1 | 51 (68.0%) | 56.8–77.5 |
| **Age groups** | 18 to 30 | 64 (55.2%) | 46.1–63.9 | 13 (37.1%) | 23.2–53.7 | 50 (62.5%) | 51.5–72.3 |
| | 31 to 60 | 55 (73.3%) | 62.4–82.0 | 4 (30.8%) | 12.7–57.6 | 38 (65.5%) | 52.7–76.4 |
| | > 61 | 0 (0.0%) | 0–79.3 | 1 (100.0%) | 20.7–100 | 1 (100.0%) | 20.7–100 |
| **Work place** | High risk | 47 (54.7%) | 44.2–64.7 | 8 (40.0%) | 21.9–61.3 | 37 (63.8%) | 50.9–74.9 |
| | Low risk | 72 (67.9%) | 58.5–76.0 | 10 (34.5%) | 19.9–52.7 | 52 (64.2%) | 53.3–73.8 |
| **Smoking** | Yes | 14 (58.3%) | 38.8–75.5 | 0 (0.0%) | 00–32.4 | 8 (42.1%) | 23.1–63.7 |
| | No | 105 (62.5%) | 55.0–69.5 | 18 (43.9%) | 29.9–59 | 81 (67.5%) | 58.7–75.2 |
| **Comorbidities** | Yes | 2 (20.0%) | 5.7–51.0 | 0 (0.0%) | 00–43.4 | 1 (14.3%) | 2.6–51.3 |
| | No | 117 (64.3%) | 57.1–70.9 | 18 (40.9%) | 27.7–55.6 | 88 (66.7%) | 58.3–74.1 |
| **Vaccination** | Not Vaccinated | 65 (53.7%) | 44.9–62.4 | 4 (12.1%) | 4.8–27.3 | 38 (49.4%) | 38.5–60.3 |
| | Partially Vaccinated | 23 (79.3%) | 61.6–90.2 | 5 (100%) | 56.6–100 | 23 (88.5%) | 71.0–96.0 |
| | Vaccinated | 31 (73.8%) | 58.9–84.7 | 9 (81.8%) | 52.3–94.9 | 28 (77.8%) | 61.9–88.3 |
| **Recent IPC training** | Not received | 20 (76.9%) | 57.9–89.0 | 2 (50.0%) | 15.0–85.0 | 76 (71.7%) | 62.5–79.4 |
| | Less than 2 hr | 89 (61.8%) | 53.7–69.3 | 16 (47.1%) | 31.5–63.3 | 1 (8.3%) | 1. 5–35.4 |
| | More than 2 hr | 10 (45.5%) | 26.9–65.3 | 0 (0.0%) | 00–25.9 | 12 (57.1%) | 36.5–75.5 |

*Percentages may not total 100 because of rounding off. HCW denotes healthcare worker, hr: hours, IPC: infection prevention control, CI: confidence interval of the percentage.

†Gender was reported by the participants

Studies done elsewhere had similar profiles of HCWs, being young, male, and paramedics but differ concerning the presence of comorbidity, smoking status, and symptoms [19, 20]. At baseline, only a few (5.7%) participants complained about symptoms. Interestingly, most of those who were seropositive did not had symptoms, similar to a countrywide serosurvey [14]. Studies across the continent found COVID-19 patients asymptomatic, with only a few requiring hospitalizations [21]. The incubation period observed was also similar to the Centers for Disease Control and Prevention [22].

In this study, HCWs were exposed to close contact exposure, prolonged face-to-face exposure, aerosol-generating procedure, exposure to patient's material, patient's body fluid, and environment surface exposure. These exposures made them a high-risk group for contracting COVID-19. Many studies demonstrate HCWs getting infected due to these procedures [23, 24]. However, we did not observe any association of type of exposure with seroconversion except among those exposed to patient material. Good adherence to PPE and IPC measures and a concurrent vaccination program could be the reason for the same. Among different categories of HCWs and types of exposures, we observed paramedics to be the least adherent to PPE, especially during low-risk activity.

A similar trend was observed among paramedics concerning adherence to appropriate hand hygiene practices. This may underline the difference in the knowledge and attitude of HCWs despite the mandatory PPE policy at the hospital. In addition to not being directly involved in patient care, paramedics may perceive they are at lower risk. We did not observe

**Table 7. Univariate regression model of variables related to seroprevalence, seroconversion, and rise in titer.**

| Variable | | Seroprevalence | | Seroconversion | | Rise in titre | |
|---|---|---|---|---|---|---|---|
| | | OR (95%CI) | p value | OR (95%CI) | p value | OR (95%CI) | p value |
| **Gender** | Female | 0.72 (0.40–1.28) | 0.26 | 1.55 (0.57–4.20) | 0.39 | 0.69 (0.34–1.38) | 0.29 |
| | Males | 1 | | 1 | | 1 | |
| **Age groups** | 30–59 | 2.23 (1.19–4.19) | 0.01 | 0.38 (0.12–1.24) | 0.11 | 1.14 (0.56–2.31) | 0.71 |
| | 60 and above | 0.00 (0.00 –NaN) | 1.000 | High (0.00 –NaN) | 1.000 | High (0.00-NaN) | 1.00 |
| | 18–29 | 1 | | 1 | | 1 | |
| **Profession** | Doctor | 0.35 (0.18–0.71) | 0.003 | 13.04 (3.39–50.25) | <0.001 | 1.40 (0.59–3.33) | 0.45 |
| | Nurse | 0.51 (0.24–1.11) | 0.09 | 3.26 (0.62–17.27) | 0.17 | 0.65 (0.27–1.61) | 0.35 |
| | Paramedic | 1 | | 1 | | 1 | |
| **Work place** | High risk | 0.57 (0.32–1.03) | 0.06 | 1.14 (0.42–3.08) | 0.80 | 0.98 (0.49–1.98) | 0.96 |
| | Low risk | 1 | | 1 | | 1 | |
| **Smoking** | Yes | 0.84 (0.35–2.00) | 0.69 | 0.00 (0.00 –NaN) | 0.99 | 0.35 (0.13–0.94) | 0.04 |
| | No | 1 | | 1 | | 1 | |
| **Comorbidities** | Yes | 0.14 (0.03–0.67) | 0.01 | 0.00 (0.00 –NaN) | 0.99 | 0.08 (0.01–0.71) | 0.02 |
| | No | 1 | | 1 | | 1 | |
| **COVID-19 symptoms** | Yes | 0.21 (0.05–0.82) | 0.025 | 0.49 (0.11–2.11) | 0.337 | 0.188 (0.06–0.64) | 0.007 |
| | No | 1 | | 1 | | 1 | |
| **Vaccination** | Vaccinated | 2.43 (1.12–5.27) | 0.03 | 6.08 (1.73–21.40) | 0.005 | 3.59 (1.46–8.87) | 0.006 |
| | Partially Vaccinated | 3.30 (1.26–8.69) | 0.02 | 4.35 (1.07–17.65) | 0.04 | 7.87 (2.18–28.40) | 0.002 |
| | Not Vaccinated | 1 | | 1 | | 1 | |
| **IPC Training** | More than 2 hr | 0.25 (0.07–0.86) | 0.03 | 0.00 (0.00 –NaN) | 0.99 | 0.07 (0.01–0.63) | 0.02 |
| | Less than 2 hr | 0.49 (0.18–1.28) | 0.15 | 1.69 (0.36–7.97) | 0.51 | 1.90 (0.73–4.97) | 0.19 |
| | Not received | 1 | | 1 | | 1 | |

*Percentages may not total 100 because of rounding off. OR: denotes odds ratio, CI: confidence interval, hr: hours

**NaN is Not a Number, unable to divide by 0.

any significant association between using individual PPE with seroconversion or an increase in titer. Studies have shown the appropriate use of PPE as the most critical defense against COVID-19 infection among HCWs [25, 26]. We could not observe this due to good PPE adherence and vaccination confounding. Unlike our study, an international study observed higher knowledge and practice of PPE in non-physicians compared to physicians [27]. Similar to ours, a study also observed that resident doctors and paramedics reported lower adherence to PPE [28]. This, along with inappropriate hand hygiene practices, could be one of the potentials for exposure to the COVID-19. Previous studies have also stressed IPC practices among various categories of HCW [29].

COVID-19 vaccination was observed to be the most important factor associated/confounded with the seropositivity and seroconversion of the HCW in this study. When we conceptualized this study, vaccination was not taken into account as it was not available globally, and was later added after protocol deviation. About two-thirds (63%) of the HCW were not vaccinated; nurses and paramedics were higher in the proportion of unvaccinated HCWs. Among the vaccinated group of HCWs, doctors were in the majority. The vaccination rates in our study were higher in comparison to the general population and healthcare workers in general, according to the national registry [30]. The United States and other developed nations had better vaccination rates; however, developing countries like India (6.2%), Brazil (16%), and Bangladesh (2.6%) had comparatively low vaccination coverage [31]. Using unvaccinated persons as a reference population, we found vaccination to be strongly and positively

associated with seropositivity (OR 2.43, 95% CI 1.12–5.27), seroconversion (OR 6.08, 95% CI 1.73–21.40) as well as serial rise in titer (OR 3.59, 95% CI 1.46–8.87). These findings support the consensus that COVID-19 vaccines produce antibodies through the immune response [32]. The seropositivity at baseline was 62%, while at endline it was 77.7%, similar to a study from Germany [13]. The latest serosurvey in India and a state-wide serosurvey in Delhi observed low seropositivity among the general population. Also, they found that doctors have a higher risk of seropositivity than other HCWs, similar to our study findings [14, 15]. Globally the seroprevalence among HCW varied from 3% in Finland, 3.3% in China, 9.3% in Spain, 10.1% in UK, 6.35–33% in the US, while metanalysis observed it to be 8.7% [7–12, 33]. Seropositivity is observed to be higher in HCWs involved in COVID-19 patient management; however, a study from Chile found no statistical difference in seropositivity among HCWs involved in the direct clinical care of patients with COVID-19 and those working in low-risk areas [20, 34]. We believe that high seroprevalence in our study population is due to the high vaccination rates, which was also one of our study's most substantial risk factors.

We also observed various factors associated with seropositivity at baseline. It was observed that being a doctor (OR 0.35, 95% CI 0.18–0.71), having symptoms (OR 0.21, 95% CI 0.05–0.82), comorbidities (OR 0.14, 95% CI 0.03–0.67), recent IPC training (OR 0.25, 95% CI 0.07–0.86), partially vaccinated (OR 3.30, 95% CI 1.26–8.69), as well as fully vaccinated for COVID-19 (OR 2.43, 95% CI 1.12–5.27) had significant risk factors for seropositivity. Age and gender of the HCW, usage of PPE, and adherence to IPC did not have a significant association with seropositivity in our study, while those with symptoms had a lower risk of seroconversion. These contrast studies from similar settings; a study from New Delhi observed seropositivity associated with the male sex [15], while WHO reported the use of PPE as protective [26]. A study from Spain reported high odds of seropositive among those having any COVID-19 symptoms in the previous months; although found no association with the profession, working in a high-risk unit, close contact with a COVID-19 case, comorbidities, and sex, partially supporting our findings [10]. We observed a higher seroprevalence in paramedical staff compared to doctors (71.6% vs. 47.1%) and those working in a low-risk area compared to a high-risk area (67.9% vs. 54.7%). This trend is similar to the adherence to PPE and IPC measures which were better among allied health workers and high-risk exposures. The healthcare setting where this study took place had a dedicated IPC committee, COVID Surveillance Unit, and proper PPE mandate, leading to better adherence to IPC measures and higher availability of PPEs compared to other hospitals and effective contact tracing.

Seroconversion was observed in 36.7% of HCWs, 63.2% in doctors, 42.9% in nurses, and 13.0% in paramedics staff. We also observed that 64.0% had an increase in the titre of antibodies during the follow-up period. Studies across the globe have varied seroconversion rates, from 0.77% in a large prospective study in the United Kingdom [7], to 77.7% in Germany after 12 week follow-up period [13]. Seroconversion among HCWs for H1N1 in 2009 was documented as 6.5% [35]. This variation could be due to different settings and study periods. High seroconversion in our study population can be attributed to the concurrent vaccination program—the most decisive risk factor observed in our study—rather than secondary infection from the COVID-19 case to which HCW was exposed. Seroconversion was positively associated with being a doctor (OR 13.04) and with partially (OR 4.35), as well as fully vaccinated for COVID-19 (OR 6.08). We also observed a negative and significant relationship of serial rise in titre of antibodies with symptoms (OR 0.17), smoking (OR 0.35), comorbidities (OR 0.08), recent IPC Training (OR 0.07), while positively associated with partially (OR 7.87), as well as fully vaccinated for COVID-19 (OR 3.59). None of the HCWs who were smokers, those with comorbidity, or those who had attended adequate IPC training, had seroconversion. We observed higher seroconversion among females, higher age groups, and those working in

high-risk units but did not reach the significance level. Our findings are supported by the various studies done in different parts of the world [20]. A negative association between smoking and seroconversion found in our study is also supported by a study from Chile, where smokers showed lower seroconversion [20]. While we observed more seroconversion in doctors, during the H1N1 epidemic, more nurses, compared to doctors, were significantly associated with seroconversion [35]. The possible explanation could be due to the confounding of vaccination against COVID-19, which was substantially higher among doctors. Thus we are of the opinion that the high seropositivity, seroconversion, and rise in titer we observed could be due to concurrent vaccination against COVID-19 rather than recent exposure to COVID-19 patients. Further study using anti-N antibodies serology may help us understand this better.

## Limitations

Despite our best efforts, we had a few limitations. The study was done during the COVID-19 pandemic, with HCWs preoccupied with their COVID duties and exhausted. Due to this, we had a high refusal and attrition rate, which could bring selection bias. In spite of a higher attrition rate, the profile of responders and those who were lost to follow-up did not vary significantly; thus attrition bias could be minimized. Due to the concurrent vaccination drive among HCWs, our study was confounded and thus prevented us from understanding the development of secondary infection among HCWs. Apart from seroconversion, utilizing RT-PCR testing could have confirmed COVID-19 infection among HCWs, but it was non-practical and unethical during the time our healthcare system was overwhelmed with the requirement of COVID testing.

## Conclusion

Our study observed higher seroprevalence among HCWs and its association with vaccination for COVID-19. The seropositivity was high among paramedical staff, but more doctors were seroconverted and had increased titer. Doctors and vaccinated HCWs were highly associated with seroconversion and protection against COVID-19. Hence, it is strongly recommended to increase vaccination coverage for all cadres of HCWs. In future research, this confounding of infection with vaccination may be curtailed using anti-N antibodies serology. It was also observed that the facility's PPE, hand hygiene, and IPC measures are practiced and protective. However, nurses and doctors had higher adherence than the paramedical staff. Therefore, imparting frequent IPC training and behavior change communication among paramedical staff is vital in preventing COVID-19 among them.

## Supporting information

**S1 File. Sample size calculation, operational Definition, participant recruitment and sample processing.**
(DOCX)

**S1 Fig. Flow diagram of participants in our study.**
(TIF)

**S2 Fig. Usage of various PPE by HCW during recent exposure to COVID-19 patients across different types of exposure.**
(TIF)

**S1 Table. Hand hygiene practiced by healthcare workers.**
(DOCX)

**S2 Table. Distribution of symptom profile with the serology at baseline.**
(DOCX)

## Acknowledgments

We are thankful to the hospital admiration of HIMSR, New Delhi, including Dr. G.N Qazi, CEO, and Dr. Ajaz Mustafa, former MS. We also would like to sincerely thank all the study participants for their time and cooperation in completing this project.

## Author Contributions

**Conceptualization:** Aqsa Shaikh, Farzana Islam, Yasir Alvi, Mohammad Ahmad.

**Data curation:** Aqsa Shaikh, Farzana Islam, Yasir Alvi, Varun Kashyap, Vishal Singh, Neetu Shree, Shyamasree Nandy, Vineet Jain.

**Formal analysis:** Farzana Islam, Yasir Alvi, Varun Kashyap, Vishal Singh.

**Funding acquisition:** Mridu Dudeja, Aqsa Shaikh, Farzana Islam, Yasir Alvi, Mohammad Ahmad, Anisur Rahman.

**Investigation:** Aqsa Shaikh, Farzana Islam, Yasir Alvi, Meely Panda, Neetu Shree, Shyamasree Nandy, Vineet Jain.

**Methodology:** Mridu Dudeja, Aqsa Shaikh, Farzana Islam, Yasir Alvi, Mohammad Ahmad, Meely Panda, Neetu Shree.

**Project administration:** Mridu Dudeja, Aqsa Shaikh, Farzana Islam, Yasir Alvi, Mohammad Ahmad, Neetu Shree, Vineet Jain.

**Resources:** Mridu Dudeja, Aqsa Shaikh, Yasir Alvi, Anisur Rahman, Meely Panda.

**Software:** Yasir Alvi, Varun Kashyap.

**Supervision:** Aqsa Shaikh, Farzana Islam, Mohammad Ahmad, Vineet Jain.

**Validation:** Aqsa Shaikh, Mohammad Ahmad, Varun Kashyap, Anisur Rahman.

**Visualization:** Varun Kashyap, Vishal Singh.

**Writing – original draft:** Aqsa Shaikh, Yasir Alvi, Vishal Singh, Meely Panda, Neetu Shree, Vineet Jain.

**Writing – review & editing:** Mridu Dudeja, Aqsa Shaikh, Farzana Islam, Yasir Alvi, Mohammad Ahmad, Vishal Singh, Anisur Rahman, Shyamasree Nandy.

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
