## [Decision Letter · Decision Letter 0]

23 Jun 2022

PONE-D-22-05833Assessment of Potential Risk Factors for COVID-19 among Health Care Workers in a Health Care Setting in Delhi, India -A Cohort StudyPLOS ONE

Dear Dr. Alvi,

Thank you for submitting your manuscript to PLOS ONE. After careful consideration, we feel that it has merit but does not fully meet PLOS ONE’s publication criteria as it currently stands. Therefore, we invite you to submit a revised version of the manuscript that addresses the points raised during the review process.

A marked-up copy of your manuscript that highlights changes made to the original version. You should upload this as a separate file labeled 'Revised Manuscript with Track Changes'.An unmarked version of your revised paper without tracked changes. You should upload this as a separate file labeled 'Manuscript'.

We look forward to receiving your revised manuscript.

Kind regards,

Amitava Mukherjee, ME, Ph.D.

Academic Editor

PLOS ONE

Journal Requirements:

Reviewers' comments:

Reviewer's Responses to Questions

**Comments to the Author**

1. Is the manuscript technically sound, and do the data support the conclusions?

Reviewer #1: Partly

2. Has the statistical analysis been performed appropriately and rigorously? 

Reviewer #1: No

3. Have the authors made all data underlying the findings in their manuscript fully available?

Reviewer #1: Yes

4. Is the manuscript presented in an intelligible fashion and written in standard English?

Reviewer #1: No

5. Review Comments to the Author

Reviewer #1: The authors present the results of a cohort study to assess potential risk factors of COVID-19 infection among healthcare workers in India.

The article needs a major revision prior being accepted.

Comments regarding the use of English language:

There are some grammatical changes required to address the English language and style. Through the whole document, the English should be reviewed to increase the readability of the document and also to sound more technical. Spelling need to be checked carefully (COIVD 19 for example). Keywords should not be acronyms. Authors should spell the complete word/term instead. Acronyms are not appropriately defined and should be reviewed. CI and OR should be reviewed to include 2 or 3 decimals consistently through the whole document. The manuscript should carefully be reviewed. Please complete a thorough proof read of the text and correct any spelling and grammar errors.

General comments:

Title should be reconsidered. Please revise the title of your manuscript to include the research question, study design and setting.

Regarding authors information, I miss information about Neetushree author as well as detailed information regarding the corresponding author.

One of the main weaknesses is the lack of updating of the bibliographic references. It is evident that the bibliography search should be reviewed and updated, because the manuscript has not considered many of the main seroprevalence studies in health workers. I would suggest performing an accurate review of similar papers already published on this topic. References should be updated. Also, bibliographic references are often poorly cited. References should be carefully reviewed.

Specific comments:

Introduction:

1. I miss a reference at the end of the second paragraph. Introduction should be more concise and add what is already published regarding infection among HCW with more detail.

2. The objective of the study is not accurate and should be better defined ant the end of the introduction section.

Material & methods:

1. There is an incongruence of the study period in the abstract and the methods section. The abstract mentions 1st and 2nd wave while methods only mention second wave. Methods should include information regarding the months in which first and second covid waves took place in the region. Please review.

2. Authors say “The study population included all the health personnel who were working in this hospital and had come in contact or been exposed recently to a COVID-19 patient receiving care” it should be stated clearly how many HCW and their working status and of whom, how many participated in the study. This information is not clear. The participants’ flowchart should be mentioned at this point.

3. I do not really see the point of calculating sample size according to the design of the study... It will be better to assess the statistical power of the analysis according to the study population.

4. More detail on how data was collected and anonymized should be provided either in “participant recruitment” or in “ethical considerations”.

5. “Pearson’s chi-square and bivariate logistic regression was done to evaluate the independent associations of multiple factors.” This analysis was also performed using excel or any statistical software was used instead? Please modify accordingly.

6. “All the participants who needed RTPCR test were offered the same by Hospital. All the participants with poor IPC practices were recommended for refresher IPC training” this should be moved to in “participant recruitment” instead of being part of “ethical considerations”.

7. I’m missing information on IRB code approval. It should be stated in the “ethical considerations” section.

Results:

1. Out of a total of 405 HCW approached, 192 were recruited in this study – please add participation rate at the end of the sentence. Results should include information on denominators 192 HCW recruited from how many? Which is the participation? This should be stated clearly in the results section but also in the abstract section.

2. Please add N= and % for each value to facilitate the comprehension of the manuscript.

3. “Out of a total of 405 HCW approached, 192 were recruited in this study. All of them were interviewed and blood sampling for serology was done at the baseline visit. Out of them, 139 were also included at end line serology assessment, reason of lost to follow up are highlighted in Figure 2.” This could be moved to methodology rather than a result. This could be part of “participant recruitment” and will be easier to read and understand.

Discussion should be more elaborated and include an update of the bibliographic references. 50.1% of participants did not provide consent to participate in the study, which is a very low participation rate. This should be stated among the limitations of the study. Also the lost to follow up has to be mentioned in limitations. Conclusions should be better presented.

Tables, figures and supplementary material:

Some principal tables should be move to supplemental, some supplemental should be move to main tables and some tables should be combined. For example: table 2 should be combined with supplemental table 1 to include all the information, as this is among the main tables of this paper, also, information on p value across different HCW groups should be added. Supplementary table 2 and 3 should be principal tables as well, and groups should be compared and statistical significances should be added.

“We observed a definite trend of adherence to PPE among various health workers as shown in Supplementary Table 2.” How this trend is evaluated? Info should be added.

Some figures need to be eliminated as they do not show relevant information. Others should be improved. Figure 1: Please review the spelling. The visualization of the figure should be improved. Footnote could be of interest. It could be supplementary material. Figure 2: Please review the spelling. It could be supplementary material. Figure 3 could be part of text only as visualization is not the most convenient. Also information regarding statistically significant differences on the text should be considered to be included. Figure 4 Visualization has to improve, numbers are very hard to read. It could be supplementary material. Figure 6 could be part of text only; the visualization does not add value to manuscript. Figure 6 information should be combined with table 3. And both should be combined with suppl table 5. Then Suppl table 5 can be a main table of results.

6. PLOS authors have the option to publish the peer review history of their article (what does this mean?). If published, this will include your full peer review and any attached files.

Reviewer #1: No

---

## [Author Response · Author response to Decision Letter 0]

26 Aug 2022

Reply of Reviewer 1 comments in detail point-by-point

 Reviewer 1 comments:R Authors reply:A

R1. Is the manuscript technically sound, and do the data support the conclusions? 

Reviewer #1: Partly 

A: We are grateful to the reviewer for providing valuable comments. We have taken this comment seriously and thoroughly revised the relevant section. We request you to go to the revised manuscript which is in track mode in MS word. You can see all the changes done in the revised manuscript.

R2. Has the statistical analysis been performed appropriately and rigorously? 

Reviewer #1: No 

A:We have revised the results section and combined the figures and tables as suggested by the reviewer. We hope the current manuscripts fills the lacunae as stated by the reviewer. 

R3. Have the authors made all data underlying the findings in their manuscript fully available? 

Reviewer #1: Yes 

A:We are grateful to you for pointing this out. 

R4. Is the manuscript presented in an intelligible fashion and written in standard English? 

Reviewer #1: No 

A:We have redrafted the manuscript and thoroughly revised the result and discussion section. We have taken the help of an external agency for improving the written English. We believe that the current version of manuscript is an improvement over the previous one.

Comments regarding the use of English language: 

R: There are some grammatical changes required to address the English language and style. Through the whole document, the English should be reviewed to increase the readability of the document and also to sound more technical. Spelling need to be checked carefully (COIVD 19 for example). Keywords should not be acronyms. Authors should spell the complete word/term instead. Acronyms are not appropriately defined and should be reviewed. 

CI and OR should be reviewed to include 2 or 3 decimals consistently through the whole document. The manuscript should carefully be reviewed. 

Please complete a thorough proof read of the text and correct any spelling and grammar errors. 

A:The manuscript has been revied by a native English speaker for correcting language and style. We have taken the help of an external agency for improving the written English. We believe that the current version of manuscript has improved in readability, clarity and is technical sound. Spelling and Acronyms have be checked and Keywords have been revised. As suggested CI and OR are made consistent. 

R:Title should be reconsidered. Please revise the title of your manuscript to include the research question, study design and setting.

Regarding authors information, I miss information about Neetushree author as well as detailed information regarding the corresponding author. One of the main weaknesses is the lack of updating of the bibliographic references. It is evident that the bibliography search should be reviewed and updated, because the manuscript has not considered many of the main seroprevalence studies in health workers. I would suggest performing an accurate review of similar papers already published on this topic. References should be updated. Also, bibliographic references are often poorly cited. References should be carefully reviewed. 

A:The title of the study is ‘Assessment of Potential Risk Factors for COVID-19 among Health Care Workers in a Health Care Setting in Delhi, India -A Cohort Study’ mentioned study design (Cohort Study) and setting (Health Care Workers in a Health Care Setting in Delhi, India). Since it is already long, we think that adding research question will make it non intuitive and decrease readability. Instead we have added the research question at the end of introduction section. 

Mis-information about one author has been corrected, as pointed out. 

With regards to the bibliographic references in introduction section, we have tried to build the rationale of our study and included those studies which were done in initial phase of pandemic, prior to designing our study. Although we have updated references in discussion section. 

Specific comments: 

Introduction

R1. I miss a reference at the end of the second paragraph. Introduction should be more concise and add what is already published regarding infection among HCW with more detail.

A: We are thankful to the reviewer for pointing out the omission of the reference. We have added it in the current version of manuscript. 

As suggested, we have worked on our introduction section and added few articles to make it concise and relevant. 

R2. The objective of the study is not accurate and should be better defined ant the end of the introduction section. 

A:We have considered your feedback and reframed the objectives to make them accurate. Hope the new manuscript is able to fulfil the objectives of the study in a better way.

Material & methods:

R1. There is an incongruence of the study period in the abstract and the methods section. The abstract mentions 1st and 2nd wave while methods only mention second wave. 

Methods should include information regarding the months in which first and second covid waves took place in the region. Please review. 

A:Thank you for pointing out errors in abstract and the methods section. It has been corrected in the revised manuscript. 

R2. Authors say “The study population included all the health personnel who were working in this hospital and had come in contact or been exposed recently to a COVID-19 patient receiving care” it should be stated clearly how many HCW and their working status and of whom, how many participated in the study. This information is not clear. The participants’ flowchart should be mentioned at this point. 

A:We think that participant enrolment needs clarity in the study population. So we have added participants’ flowchart in the methods section. 

R3. I do not really see the point of calculating sample size according to the design of the study... It will be better to assess the statistical power of the analysis according to the study population.

A:Since this was a Cohort Study, we observed and followed only those cases that had an exposure to COVID-19 patients. While designing the study, we didn’t plan to observe any unexposed groups. 

For power analysis, we needed the data of exposed and unexposed group to test for development of antibodies against COVID-19, so we could not assess as the reviewer had suggested. But since there was some confusion regarding sample size in the previous version of manuscript, we have now calculated the sample size using population proportion formula to make it more clear. 

R4. More detail on how data was collected and anonymized should be provided either in “participant recruitment” or in “ethical considerations”.

A:We have rewritten the Participant Recruitment section in more detail.

R5. “Pearson’s chi-square and bivariate logistic regression was done to evaluate the independent associations of multiple factors.” This analysis was also performed using excel or any statistical software was used instead? Please modify accordingly. 

A:All the statistical analysis were performed using IBM SPSS version 26. We have rewritten the statistical analyses section to make it clear to all.

R6. “All the participants who needed RTPCR test were offered the same by Hospital. All the participants with poor IPC practices were recommended for refresher IPC training” this should be moved to in “participant recruitment” instead of being part of “ethical considerations”. 

A:We have shifted these sentences as suggested by the reviewer 

R7. I’m missing information on IRB code approval. It should be stated in the “ethical considerations” section. 

A:Thank you for pointing this out. We have added the IEC code in the current version of manuscript. 

Results:

R1. Out of a total of 405 HCW approached, 192 were recruited in this study – please add participation rate at the end of the sentence. Results should include information on denominators 192 HCW recruited from how many? Which is the participation? This should be stated clearly in the results section but also in the abstract section.

A:We want to clarify about enrolment. 

Only 405 HCWs had recent exposure to COVID-19 patients and were our potential participants who could be recruited (sample frame). They were screened with the inclusion and exclusion criteria, and 192 were recruited in this study. After excluding 10 participant who were not eligible, 192/395 is the participation rate. 

R2. Please add N= and % for each value to facilitate the comprehension of the manuscript. 

A:We have provided Frequency (N) and percentage (%) of every variable in the tables, but avoided this information in text as it would be duplication of information in the tables.

R3. “Out of a total of 405 HCW approached, 192 were recruited in this study. All of them were interviewed and blood sampling for serology was done at the baseline visit. Out of them, 139 were also included at end line serology assessment, reason of lost to follow up are highlighted in Figure 2.” This could be moved to methodology rather than a result. This could be part of “participant recruitment” and will be easier to read and understand. 

A:Thank you for your suggestion and we have shifted the information in method section but have revised them in result section for comprehension to the readers

Discussion 

R: Discussion should be more elaborated and include an update of the bibliographic references. 

50.1% of participants did not provide consent to participate in the study, which is a very low participation rate. This should be stated among the limitations of the study. Also the lost to follow up has to be mentioned in limitations. 

Conclusions should be better presented. 

A: Thankyou for the feedback. We have revised the discussion and updated references. 

Yes, we had this was a limitation, but major reason for this was a lack of time with doctors and nurses, as they were exhausted with their covid duty and were not able to provide time for this study. As suggested we have added these in limitations. 

We have redrafted conclusion to make it clearer

R: Tables, figures and supplementary material: Some principal tables should be move to supplemental, some supplemental should be move to main tables and some tables should be combined. 

For example: table 2 should be combined with supplemental table 1 to include all the information, as this is among the main tables of this paper, also, information on p value across different HCW groups should be added. 

Supplementary table 2 and 3 should be principal tables as well, and groups should be compared and statistical significances should be added.

“We observed a definite trend of adherence to PPE among various health workers as shown in Supplementary Table 2.” How this trend is evaluated? Info should be added.

Some figures need to be eliminated as they do not show relevant information. Others should be improved. 

Figure 1: Please review the spelling. The visualization of the figure should be improved. Footnote could be of interest. It could be supplementary material. 

Figure 2: Please review the spelling. It could be supplementary material. 

Figure 3 could be part of text only as visualization is not the most convenient. Also information regarding statistically significant differences on the text should be considered to be included. 

Figure 4 Visualization has to improve, numbers are very hard to read. It could be supplementary material. 

Figure 6 could be part of text only; the visualization does not add value to manuscript. Figure 6 information should be combined with table 3. And both should be combined with suppl table 5. Then Suppl table 5 can be a main table of results. 

A:Thankyou for the important suggestions. 

We have merdge the table 2 with supplemental table 1, although we could not analyse the association as we lack sufficient samples for each category.. 

Supplementary table 2 and 3 are labled as principal tables as suggested, although we calculated the statistical significances for Supplementary table 2, but did not have enough sample size for assessing association of individual subpopulation of HCWs for Supplementary table 3.

Figure 1 is made supplementary Figure 1 along with spell check and footnote

Figure 2 is now figure 1 and checked for spelling. We have added more information and now show participants enrolment and retained as principal figure as you have suggested in methods section. 

Figure 3 is now figure 2. We have used percentages of the categories for more clarity and calculated significance, as suggested.

Figure 4 is now supplementary figure 2 and we have tried to improve the visualization to improve clarity.

As suggested we have merged the information from figure 6, table 3 and table 5 and prepared new table (Table 6)

The authors have redrafted the whole manuscript. The final version that we are now submitting is after getting edited by professional editing service.

---

## [Decision Letter · Decision Letter 1]

17 Oct 2022

PONE-D-22-05833R1Assessment of Potential Risk Factors for COVID-19 among Health Care Workers in a Health Care Setting in Delhi, India -A Cohort StudyPLOS ONE

Dear Dr. Alvi,

Thank you for submitting your manuscript to PLOS ONE. After careful consideration, we feel that it has merit but does not fully meet PLOS ONE’s publication criteria as it currently stands. Therefore, we invite you to submit a revised version of the manuscript that addresses the points raised during the review process.

ACADEMIC EDITOR: Please pay serious attention to the comments of the reviewer and revise your manuscript accordingly. 

We look forward to receiving your revised manuscript.

Kind regards,

Amitava Mukherjee, ME, Ph.D.

Academic Editor

PLOS ONE

Reviewers' comments:

Reviewer's Responses to Questions

**Comments to the Author**

1. If the authors have adequately addressed your comments raised in a previous round of review and you feel that this manuscript is now acceptable for publication, you may indicate that here to bypass the “Comments to the Author” section, enter your conflict of interest statement in the “Confidential to Editor” section, and submit your "Accept" recommendation.

Reviewer #1: (No Response)

2. Is the manuscript technically sound, and do the data support the conclusions?

Reviewer #1: Partly

3. Has the statistical analysis been performed appropriately and rigorously? 

Reviewer #1: Yes

4. Have the authors made all data underlying the findings in their manuscript fully available?

Reviewer #1: Yes

5. Is the manuscript presented in an intelligible fashion and written in standard English?

Reviewer #1: Yes

6. Review Comments to the Author

Reviewer #1: General comments:

The authors have addressed the comments sufficiently. Nevertheless, the manuscript needs some improvements prior to publication.

Despite the authors’ mention that a professional English reviewer/editor has reviewed the manuscript, some spelling mistakes are still observed in the manuscript, as well as conceptual errors (use of COVID-19 to refer to the virus, not for the disease), and readability still needs to be improved. Some examples:

- 1st paragraph introduction: “COVID-19 belongs to the large family of viruses Coronaviridae, the viruses have the peculiar property of constantly changing and becoming diverse, which has helped them spread and survive and jeopardize any health system.” It should read (for example): “SARS-CoV-2 virus belongs to the large family of viruses Coronaviridae. This virus has constantly been changing and becoming diverse, and this property has helped this virus to spread among vulnerable populations and jeopardize any health system quickly”. – like this edit, many parts of the manuscript could be improved

- 1st paragraph introduction: Coronaviridae needs to be in italics

- 1st and second sentences in the 1st paragraph of the introduction: Use SARS-CoV-2 to refer to the virus and COVID-19 to refer to the disease (example: SARS-CoV-2 virus belongs… / SARS-CoV-2 transmits from… / the rest of the paragraph using COVID is correct).

The use of acronyms needs revision as well.

- HCW appear for the first time in the second paragraph of the introduction, and the acronym is not explained (HCW providing COVID-19 care are at increased… - Should read as: “Healthcare workers (HCW) providing COVID-19 care are at increased…”; please check for consistency on other acronyms.

Also, some statistical analysis and data visualization in tables and figures need to be improved. P. ex. You should check the use of Mean ± SD and Median (IQR), as it depends on the type of variable and distribution that one is better used than the other. Please review and modify the manuscript accordingly. Further details are provided below.

Please review the acronyms as there are still mistakes: for example: confidence intervals are written entirely although acronyms defined, etc. Some acronyms are defined but only used once (HICC, HAIs, HAHC, RPAC). Please check carefully.

References still need to be reviewed; please consider ENE COVID Study or other HCW seroprevalences studies in Spain, for example.

Specific comments:

Introduction

1. “India has the largest number of confirmed cases in Asia and the second-highest number of confirmed cases in the world” – state the period: the whole pandemic? 1st wave? 2nd? 1st and 2nd?

2. Regarding the study objectives, the way they are written in the introduction does not sound adequate for a scientific article “So, the key research question we want to explore in this study is: Does providing COVID care make HCW vulnerable to COVID-19 infection compared to the general population? If yes, what are its risk factors? We did this study with the objectives (1) to understand the extent of human-to-human transmission and to find out risk factors for COVID19 among HCW in recent contact with a COVID-19 patient; and (2) to evaluate the effectiveness of infection prevention and control (IPC) measures among HCW.” I consider it should be reviewed and consider re-writing the whole last part of the last paragraph in the introduction as: “The serological data among Indian HCWs is limited. A countrywide serosurvey after the first wave observed seroprevalence among HCW at 25.6%, while it was 12.1% among HCW from teaching hospital in Delhi (14,15). Investigating the serological response and assessing the potential risk factors among health workers may help characterize virus transmission patterns, prevent future infections of health workers, and prevent the healthcare-associated spread of COVID-19. Thus, this study aims to understand the extent of human-to-human transmission and to find out risk factors for COVID-19 among Indian HCW and to evaluate the effectiveness of IPC measures among them.” – please note IPC acronym has already been defined in the previous paragraph

Methods

Sample size:

1. Based on extensive. The article is missing p.ex: based on an extensive…

2. The formula can be in the supplementary material, although I consider that it is not needed. If excluded, please review the use of the acronyms in that paragraph.

Participant recruitment:

1. When a confirmed case of COVID-19 was admitted or detected, all its suspected contact were traced by the Covid Surveillance Unit. –> contacts in plural?

2. those eligible were invited and provided a Patient Information Sheet explaining the study. -> those eligible were invited to participate in the study and a Patient Information Sheet explaining the study was provided to all of them.

Serology assessment: please check acronyms as Cut-off value (C.O.) is stated twice.

Statistical analysis:

1. p-value less than 0.05 -> p-value <0.05

2. The categorical variables were presented as percentages (%) while mean/median and standard deviation/ interquartile range (IQR) were calculated for continuous variables à it must be clearer were mean and median are used as well as when SD or IQR; also, IQR acronym was defined previously, please check.

In the ethical considerations section, I still miss the compliance of any international guideline/code of practice/ legislation. This needs to be clarified.

Results

1. (participation rate 192/395) consider %

2. Of them, 139 were also  add %

3. The lowest percentage of unvaccinated individuals (121/192) was among doctors (47.1%), while paramedics had the lowest rate (9.8%) among fully vaccinated, and this difference was statistically significant (Figure 2). à consider wording/modify (suggestion: “The lowest percentage of unvaccinated individuals was among doctors (N=XXX, 47.1%), while paramedics had the lowest rate (9.8%) of fully vaccinated individuals (p<0.001) (Figure 2).).

...among them, 27 had prolonged face-to-face exposure, and 10 had exposure during the aerosol generating procedure, while exposure with body fluid was observed in 10.7%. -- use N or % or both N(%) or % (N/N); but be consistent.

4. We also assessed the risk of seropositivity, seroconversion, and rise in titre as highlighted in table 7. The seropositivity was significantly and negatively associated with doctor as profession [Odds Ratio (OR):0.35, 95% Confidence Interval (CI):0.18-0.71], COVID-19 symptoms [OR:0.21, CI:0.05-0.82], comorbidities [OR:0.14, CI: 0.03 - 0.67], and IPC training [OR:0.25, CI:0.07 - 0.86], while positively associated with middle age [OR 2.23, CI:1.19 -4.19], partially [OR:3.30, CI: 1.26-8.69], as well as fully vaccination for COVID-19 [OR:2.43, CI:1.12-5.27].  please be consistent with how results are written and exposed and review the use of previously defined acronyms.

Discussion:

1. Please be consistent with how results (OR and CI) are written and review the use of previously defined acronyms.

Some parts of the discussion could be better developed, although the authors have tried to improve this section. The explanation for the results should be better performed in different parts of the discussion and also increase the comparison with previously published studies.

2. The way the discussion is started does not seem appropriate; the wording of the 1st paragraph should be reviewed.

Limitations

1. Because of lack of time, we had a high refusal rate to participate, especially by doctors and nurses, as they were exhausted in their covid duty, which could bring selection bias. à the lack of time of who? Rewrite. Covid or pandemic duty?

2. We also had a higher attrition rate than expected, although the profile of responders and non-responders did not vary significantly. -> please refer to this information regarding the significance and please try to explain why this could be.

3. Confirmation with RT-PCR testing might have helped  to what? It does not read clear.

Conclusion & policy consideration

Conclusions could improve.

1. Consider maintain “conclusions" as heading, I think it is clearer.

2. We are in opinion that it would be appropriate to regularly test all healthcare workers for COVID-19, using both PCR and serological assays, irrespective of exposure or symptom history to protect this workforce.  please re-write it is not technically sound enough.

Tables and Figures

- COVID-19 denotes coronavirus disease 2019  does not need to be in the foot tables as it is widely used, and has not been explained this acronym in the manuscript

- Table 1: decide which is better to describe the “Incubation period”: Mean ± SD or Median (IQR). It does not make sense to use both. Please see previous comments regarding this.

- Table 2: add significance (p-value) and a footnote explaining the test used for calculation; Direct Exposure -> exposure does not need capital letters.

- Table 3: column title can be p-value instead of significance; please be consistent in the different tables. A footnote must be added in the table to specify the test used for calculation.

- Table 4: A footnote needs to be added to specify the test used for calculation/ .010 à missing a “0”  0.010 / Prolong face to face à prolonged? /# Prolong contact was considered when it was >15min face to face  prolonged?

- Table 5: Adq: adequate  Adequate? / 3 (60%) 2 (40%) à consistency, always 1 decimal? Please check the whole manuscript / .759 - .188  consistency add 0.XXX? Please check the whole manuscript.

- Table 6 I would suggest not adding 95%CI but instead p-values comparing the different groups. Please, consider. Also, the 95%CI refers to the % and needs to be clarified if maintained.

- Table 7: please be consistent and put the reference category always the first or the last to make it easy for the reader to interpret the table and the results.

- Figure 1: % should be added in all the cases.

7. PLOS authors have the option to publish the peer review history of their article (what does this mean?). If published, this will include your full peer review and any attached files.

Reviewer #1: No

---

## [Author Response · Author response to Decision Letter 1]

20 Nov 2022

Reviewer 1 comments Authors reply

R:1. If the authors have adequately addressed your comments raised in a previous round of review and you feel that this manuscript is now acceptable for publication, you may indicate that here to bypass the “Comments to the Author” section, enter your conflict of interest statement in the “Confidential to Editor” section, and submit your "Accept" recommendation. 

Reviewer #1: (No Response) 

R:2. Is the manuscript technically sound, and do the data support the conclusions? 

Reviewer #1: Partly 

A: We are grateful to the reviewer for providing valuable comments. We have taken comments seriously and thoroughly revised the relevant section. We request you to go to the revised manuscript which is in track mode in MS word. You can see all the changes done in the revised manuscript.

 R:3. Has the statistical analysis been performed appropriately and rigorously? 

 Reviewer #1 Yes

A: We are happy that reviewer now considered 

statistical analysis done in the revised manuscript appropriate and rigorous. We hope all other lacunas as mentioned by them would be fulfilled in current version of manuscript. 

 R:4. Have the authors made all data underlying the findings in their manuscript fully available? 

Reviewer #1: Yes 

A: We are grateful to you for pointing this out. 

 R:5. Is the manuscript presented in an intelligible fashion and written in standard English? 

Reviewer #1: yes

A: We are happy that reviewer now considered the revised manuscript presented in intelligible fashion. We have further improved the English as suggested by the reviewer. We are grateful to the reviewer for the overall improvement in the manuscript.

General comments: R:The authors have addressed the comments sufficiently. Nevertheless, the manuscript needs some improvements prior to publication. 

Despite the authors’ mention that a professional English reviewer/editor has reviewed the manuscript, some spelling mistakes are still observed in the manuscript, as well as conceptual errors (use of COVID-19 to refer to the virus, not for the disease), and readability still needs to be improved. Some examples: 

- 1st paragraph introduction: “COVID-19 belongs to the large family of viruses Coronaviridae, the viruses have the peculiar property of constantly changing and becoming diverse, which has helped them spread and survive and jeopardize any health system.” It should read (for example): “SARS-CoV-2 virus belongs to the large family of viruses Coronaviridae. This virus has constantly been changing and becoming diverse, and this property has helped this virus to spread among vulnerable populations and jeopardize any health system quickly”. – like this edit, many parts of the manuscript could be improved 

A: The authors have further improved the grammar and conceptual errors as suggested by the reviewer, to improve its readability, clarity and make it technical sound. 

- Corrections done. Thankyou for pointing out the error

 R:- 1st paragraph introduction: Coronaviridae needs to be in italics 

A: Correction done. Thankyou for

 R: 1st and second sentences in the 1st paragraph of the introduction: Use SARS-CoV-2 to refer to the virus and COVID-19 to refer to the disease (example: SARS-CoV-2 virus belongs... / SARS-CoV-2 transmits from... / the rest of the paragraph using COVID is correct).

A: correction done. Thankyou for pointing out the error

 R:The use of acronyms needs revision as well.

- HCW appear for the first time in the second paragraph of the introduction, and the acronym is not explained (HCW providing COVID-19 care are at increased... - Should read as: “Healthcare workers (HCW) providing COVID-19 care are at increased...”; please check for consistency on other acronyms. 

A: correction done. Thankyou for pointing out the error

 R:Also, some statistical analysis and data visualization in tables and figures need to be improved. P. ex.You should check the use of Mean ± SD and Median (IQR), as it depends on the type of variable and distribution that one is better used than the other. Please review and modify the manuscript accordingly. Further details are provided below.

A: We are thankful to the reviewer for pointing out the error. We have corrected it in the current version of manuscript. 

 R:Please review the acronyms as there are still mistakes: for example: confidence intervals are written entirely although acronyms defined, etc. Some acronyms are defined but only used once (HICC, HAIs, HAHC, RPAC). Please check carefully.

A: correction done. Thankyou for suggestion

 R:References still need to be reviewed; please consider ENE COVID Study or other HCW seroprevalences studies in Spain, for example. 

A: Thankyou for suggestion. The ENE COVID study is a population-based sero-epidemiological study from Spain.

In our manuscript we have tried to included studies from different countries done among HCW and one study at Spanish hospitals is included (Garcia-Basteiro et al). 

Specific comments: 

Introduction 

 R:1. “India has the largest number of confirmed cases in Asia and the second-highest number of confirmed cases in the world” – state the period: the whole pandemic? 1st wave? 2nd? 1st and 2nd?

A: This statement is for whole pandemic. For clarity, we have added ‘till date’ in the sentence. 

 R:2. Regarding the study objectives, the way they are written in the introduction does not sound adequate for a scientific article “So, the key research question we want to explore in this study is: Does providing COVID care make HCW vulnerable to COVID-19 infection compared to the general population? If yes, what are its risk factors? We did this study with the objectives (1) to understand the extent of human-to-human transmission and to find out risk factors for COVID19 among HCW in recent contact with a COVID-19 patient; and (2) to evaluate the effectiveness of infection prevention and control (IPC) measures among HCW.” I consider it should be reviewed and consider re-writing the whole last part of the last paragraph in the introduction as: “The serological data among Indian HCWs is limited. A countrywide serosurvey after the first wave observed seroprevalence among HCW at 25.6%, while it was 12.1% among HCW from teaching hospital in Delhi (14,15). Investigating the serological response and assessing the potential risk factors among health workers may help characterize virus transmission patterns, prevent future infections of health workers, and prevent the healthcare-associated spread of COVID-19. Thus, this study aims to understand the extent of human-to- human transmission and to find out risk factors for COVID-19 among Indian HCW and to evaluate the effectiveness of IPC measures among them.” – please note IPC acronym has already been defined in the previous paragraph 

A: We are thankful to the reviewer for such an elaborated and useful suggestion. We agree with the reviewer that research question is not intuitive and has been removed in the current manuscript. We also have reframed the objectives as suggested. 

Methods 

 R:Sample size:

1. Based on extensive. The article is missing p.ex: based on an extensive...

2. The formula can be in the supplementary material, although I consider that it is not needed. If excluded, please review the use of the acronyms in that paragraph. 

A: We have added reference of the study based on which we estimated the sample sise. Also, the formula is moved to supplementary material, as suggested. 

 R:Participant recruitment:

1. When a confirmed case of COVID-19 was admitted or detected, all its suspected contact were traced by the Covid Surveillance Unit. –> contacts in plural?

A: We are thankful to the reviewer for pointing out the error. We have corrected it in the current version of manuscript. 

 R:2. those eligible were invited and provided a Patient Information Sheet explaining the study. -> those eligible were invited to participate in the study and a Patient Information Sheet explaining the study was provided to all of them. 

A: We are thankful to the reviewer for pointing out the error. We have corrected it in the current version of manuscript. 

 R:Serology assessment: please check acronyms as Cut-off value (C.O.) is stated twice. 

A: Acronyms are now properly used in the current manuscript. Thank you for pointing out. 

 R:Statistical analysis:

1. p-value less than 0.05 -> p-value <0.05 

A: We are thankful to the reviewer for pointing out the error. We have corrected it in the current version of manuscript. 

 R:2. The categorical variables were presented as percentages (%) while mean/median and standard deviation/ interquartile range (IQR) were calculated for continuous variables à it must be clearer were mean and median are used as well as when SD or IQR; also, IQR acronym was defined previously, please check

A: We have revised the sentence adding clarity. IQR is used in Statistical analyses section first in the manuscript. 

 R:In the ethical considerations section, I still miss the compliance of any international guideline/code of practice/ legislation. This needs to be clarified.

A: Thankyou for pointing this out. We have followed Guidelines of International Conference on harmonisation – Good clinical practices and added a sentence in ethical considerations section. 

Results R:1. (participation rate 192/395) consider % 2. Of them, 139 were also  add %

A: Both percentages added as suggested

 R:3. The lowest percentage of unvaccinated individuals (121/192) was among doctors (47.1%), while paramedics had the lowest rate (9.8%) among fully vaccinated, and this difference was statistically significant (Figure 2). 

à consider wording/modify (suggestion: “The lowest percentage of unvaccinated individuals was among doctors (N=121, 47.1%), while paramedics had the lowest rate (9.8%) of fully vaccinated individuals (p<0.001) (Figure 2).). 

...among them, 27 had prolonged face-to-face exposure, and 10 had exposure during the aerosol generating procedure, while exposure with body fluid was observed in 10.7%. -- use N or % or both N(%) or % (N/N); but be consistent.

A: Thankyou for the useful suggestions. We have revised the result section using N(%) consistently. 

 R:4. We also assessed the risk of seropositivity, seroconversion, and rise in titre as highlighted in table 7. The seropositivity was significantly and negatively associated with doctor as profession [Odds Ratio (OR):0.35, 95% Confidence Interval (CI):0.18-0.71], COVID-19 symptoms [OR:0.21, CI:0.05-0.82], comorbidities [OR:0.14, CI: 0.03 - 0.67], and IPC training [OR:0.25, CI:0.07 - 0.86], while positively associated with middle age [OR 2.23, CI:1.19 -4.19], partially [OR:3.30, CI: 1.26-8.69], as well as fully vaccination for COVID-19 [OR:2.43, CI:1.12-5.27].  please be consistent with how results are written and exposed and review the use of previously defined acronyms. 

A: We have used Odds Ratio (OR) and Confidence Interval (CI) acronyms in abstract before results section. As suggested, now we have removed them from results section. 

Discussion: 

 R:1. Please be consistent with how results (OR and CI) are written and review the use of previously defined acronyms. 

Some parts of the discussion could be better developed, although the authors have tried to improve this section. The explanation for the results should be better performed in different parts of the discussion and also increase the comparison with previously published studies. 

A: We are thankful to the reviewer for pointing this inconsistency. In the current manuscript we have revised as per recommendation. 

 R:2. The way the discussion is started does not seem appropriate; the wording of the 1st paragraph should be reviewed. 

A: Your suggestion have been incorporated in the revised manuscript and we have redraft the first paragraph of discussion section.

Limitations 

 R:1. Because of lack of time, we had a high refusal rate to participate, especially by doctors and nurses, as they were exhausted in their covid duty, which could bring selection bias. 

à the lack of time of who? Rewrite. Covid or pandemic duty?

2. We also had a higher attrition rate than expected, although the profile of responders and non-responders did not vary significantly. -> please refer to this information regarding the significance and please try to explain why this could be. 

3. Confirmation with RT-PCR testing might have helped  to what? It does not read clear.

A: Yes, the many HCW participants prospects did not gave consent stating lack of time after their COVID duties. 

We have considered your feedback and reframed the limitation section to make it clear. Thankyou. 

Conclusion & policy consideration 

 R:Conclusions could improve.

1. Consider maintain “conclusions" as heading, I think it is clearer.

2. We are in opinion that it would be appropriate to regularly test all healthcare workers for COVID-19, using both PCR and serological assays, irrespective of exposure or symptom history to protect this workforce.  please re-write it is not technically sound enough. 

A: We had used Conclusion & policy consideration as it was recommend in IJID guidelines. As suggested we have revised it now. 

We have revised the conclusion section as per your suggestion. We are thankful for the same. 

Tables and Figures 

 R:- COVID-19 denotes coronavirus disease 2019  does not need to be in the foot tables as it is widely used, and has not been explained this acronym in the manuscript

- Table 1: decide which is better to describe the “Incubation period”: Mean ± SD or Median (IQR). It does not make sense to use both. Please see previous comments regarding this. 

A: Thank you for your suggestion. We have incorporated both of them in the revised manuscript. For describing Incubation period in our study have used Median and IQR as the data was not normal. 

 R:- Table 2: add significance (p-value) and a footnote explaining the test used for calculation; Direct Exposure -> exposure does not need capital letters.

- Table 3: column title can be p-value instead of significance; please be consistent in the different tables. A footnote must be added in the table to specify the test used for calculation. 

- Table 4: A footnote needs to be added to specify the test used for calculation/ .010 à missing a “0”  0.010 / Prolong face to face à prolonged? /# Prolong contact was considered when it was >15min face to face  prolonged?

- Table 5: Adq: adequate  Adequate? / 3 (60%) 2 (40%) à consistency, always 1 decimal? Please check the whole manuscript / .759 - .188  consistency add 0.XXX? Please check the whole manuscript. 

A: As suggested, we have added the significance for each variable. We have rewritten the footnote specify the test used and comments raised by reviewer. 

We have used p value to three decimal places and percentage to one decimal placed in the table.

 R:- Table 6 I would suggest not adding 95%CI but instead p-values comparing the different groups. Please, consider. Also, the 95%CI refers to the % and needs to be clarified if maintained.

A: We are in opinion that 95% CI of the variable is beneficial for the manuscript, thus have maintain. We have added clarification in footnote as suggested by the reviewer. 

 R:- Table 7: please be consistent and put the reference category always the first or the last to make it easy for the reader 

A: Thank you for pointing this out. We have made last category of all the variable as reference category throughout the table. The authors have redrafted the whole manuscript. The final version that we are now submitting is after getting edited by professional editing service.

---

## [Editor Report · Decision Letter 2]

29 Nov 2022

Assessment of Potential Risk Factors for COVID-19 among Health Care Workers in a Health Care Setting in Delhi, India -A Cohort Study

PONE-D-22-05833R2

Dear Dr. Alvi,

We’re pleased to inform you that your manuscript has been judged scientifically suitable for publication and will be formally accepted for publication once it meets all outstanding technical requirements.

Kind regards,

Amitava Mukherjee, ME, Ph.D.

Academic Editor

PLOS ONE
---

## [Editor Report · Acceptance letter]

9 Dec 2022

PONE-D-22-05833R2 

Assessment of Potential Risk Factors for COVID-19 among Health Care Workers in a Health Care Setting in Delhi, India -A Cohort Study 

Dear Dr. Alvi:

I'm pleased to inform you that your manuscript has been deemed suitable for publication in PLOS ONE. Congratulations! Your manuscript is now with our production department. 

Kind regards, 

on behalf of

Professor Dr. Amitava Mukherjee 

Academic Editor

PLOS ONE